# Investigation to Explain Bioequivalence Failure in Pravastatin Immediate-Release Products

**DOI:** 10.3390/pharmaceutics11120663

**Published:** 2019-12-09

**Authors:** Alejandro Ruiz-Picazo, Sarin Colón-Useche, Blanca Perez-Amorós, Marta González-Álvarez, Irene Molina-Martínez, Isabel González-Álvarez, Alfredo García-Arieta, Marival Bermejo

**Affiliations:** 1Engineering, Pharmacokinetics and Pharmaceutical Technology Area, Miguel Hernandez University, 03550 San Juan de Alicante, Spain; alejandroruizpicazo@gmail.com (A.R.-P.); saringabriela@yahoo.es (S.C.-U.); bperezamoros@gmail.com (B.P.-A.); marta.gonzalez@goumh.umh.es (M.G.-Á.); mbermejo@goumh.umh.es (M.B.); 2Pharmacokinetics and Pharmaceutical Technology, Complutense University of Madrid, 28040 Madrid, Spain; iremm@farm.ucm.es; 3Analysis and Control Department, University of Los Andes, Merida 05101, Venezuela; 4Service on Pharmacokinetics and Generic Medicines, Division of Pharmacology and Clinical Evaluation, Department of Human Use Medicines, Spanish Agency for Medicines and Health Care Products, 28022 Madrid, Spain; agarciaa@aemps.es

**Keywords:** bioequivalence, Biopharmaceutics Classification System, in vitro, dissolution test, pravastatin

## Abstract

The purpose of this work is to explore the predictive ability of the biopharmaceutics classification system (BCS) biowaiver based on the dissolution methods for two pravastatin test products, where one of them showed bioequivalence (BE) while the other test failed (non-bioequivalence, or NBE), and to explore the reasons for the BE failure. Experimental solubility and permeability data confirmed that pravastatin is a BCS class III compound. The permeability experiments confirmed that the NBE formulation significantly increased pravastatin permeability, and could explain its higher absorption rate and higher C_max_. This finding highlights the relevance of requiring similar excipients for BCS class III drugs. The BCS-based biowaiver dissolution tests at pH 1.2, 4.5, and 6.8, with the paddle apparatus at 50 rpm in 900 mL media, were not able to detect differences in pravastatin products, although the NBE formulation exhibited a more rapid dissolution at earlier sampling times. Dissolution tests conducted in 500 mL did not achieve complete dissolution, and both formulations were dissimilar because the amount dissolved at 15 min was less than 85%. The difference was less than 10% at pH 1.2 and 4.5, while at pH 6.8 *f*_2_, results reflected the C_max_ rank order.

## 1. Introduction

The scientific rationale for accepting biopharmaceutics classification system (BCS) biowaivers and in vitro demonstrations of bioequivalence (BE) is based on the assumption that drug permeability and solubility as classification parameters are the limiting factors for drug absorption, and that the excipients contained in the drug products do not affect intestinal permeability or motility (such as gastric emptying or intestinal transit time). For products containing class III (high-solubility, and low-permeability) drugs, which demonstrate very rapid (>85% in 15 min) and similar in vitro dissolutions to that of the reference product at all physiological pHs, it is assumed that test and reference products behave as drug solutions once emptied from the stomach into the duodenum, and consequently their bioavailabilities in rate and extent must be similar if excipients do not alter the drug absorptions. In vitro BE or biowaivers based on the BCS are now included in the main regulatory guidances around the world, with some slight discrepancies on classification boundaries summarized and discussed by Lenic et al. [1], Zheng et al. [2], and Hoffsäss and Dressman [3].

Two recent reports estimated the probability of proving BE (or the risk of obtaining non-bioequivalent (NBE) or bio-inequivalent (BI) results) for products containing drugs from all BCS classes, and whether the quality control (QC) in vitro dissolution test could predict the in vivo bioequivalence outcome [4,5].

Ramirez et al. published in 2010 a survey of 124 bioequivalence studies of drugs that were classified according to the BCS in order to explore the probability of passing the BE standard for the different BCS classes and the predictive ability of the quality control dissolution test of the BE outcome [5]. In the survey, they found several drug products (including pravastatin) that failed the BE demonstration, in spite of the adequate power of the BE study design (>80%) and the fact that the drug products passed the QC dissolution test. In other words, even if the study had enough power to correctly conclude bioequivalence (H1 alternative hypothesis), the products were found to be inequivalent, and the null hypothesis (H0) of inequivalence could not be rejected [6,7]. The authors concluded that the usually employed QC dissolution tests were not adequate to allow a biowaiver of in vivo bioequivalence studies.

In the Cristofoletti et al. report [4], the authors surveyed a random sample of 500 BE studies from a database from the Brazilian medicines agency (ANVISA). In this study, the drugs were classified according to the BCS and Biopharmaceutical Drug Disposition and Classification System (BDDCS) to evaluate the outcome of bioequivalence studies. For Cristofoletti et al., the failure in pravastatin (a class III compound), in spite of the adequate power of the studies (>80%), remains unexplained.

In both surveys, the probability of obtaining a BE result when the dissolution profiles were similar was around 90% for class I and III drug products (post-test probability or positive predictive value), whereas for class II drug products, the post-test BE probability after a similar dissolution profile was 61%. On the other hand, the probability of false positive results (i.e., similar dissolution but NBE results) was almost 90% for class II drugs. These results point out the lack of an in vivo predictive value of the pharmacopeia dissolution tests [4,5].

The purpose of this work is to explore the predictive ability of BCS biowaiver-based dissolution methods for two test pravastatin products versus the innovator reference product, where one of the test products corresponds to the failing product from Ramirez et al. survey [5], and to explore the reasons for the BE failure. In addition, we have investigated the influence of the product excipients on intestinal permeability to assess the need of additional in vitro tests (apart from dissolution) to ensure the in vitro predictability of the in vivo bioequivalence outcome.

## 2. Materials and Methods

### 2.1. Compounds

Pravastatin (MW = 446.52 g/mol) was obtained from Sigma-Aldrich (Barcelona, Spain), and Acetonitrile (ACN) was obtained from VWR International (West Chester, PA, USA). Methanol (MeOH), hydrogen chloride (HCl), and trifluoroacetic acid (TFA) were purchased from Fisher Scientific (Pittsburgh, PA, USA). Sodium hydroxide (NaOH), sodium chloride (NaCl), and sodium dihydrogen phosphate monohydrate (NaH_2_PO_4_·H_2_O) were received from Sigma-Aldrich (Barcelona, Spain). Pravastatin is a weak acid with a p*K*_a_ = 4.21 and log *P* = 1.65 [8]. For oral administration, it is used in the form of sodium salt. Pravastatin sodium is a white hygroscopic powder, easily soluble in water and methanol, and acetonitrile, and practically insoluble in chloroform [9].

Metoprolol, *n*-octanol, acetonitrile, and methanol were purchased from Sigma (Barcelona, Spain).

### 2.2. Pravastatin Formulations

Lipemol 40 mg tablets (Bristol-Myers, Squibb, S.A., London, UK) were used as reference products. Their excipients are croscarmellose sodium, magnesium stearate, magnesium oxide, microcrystalline cellulose, yellow iron oxide, anhydrous lactose, and povidone K30.

Pravastatin bioequivalent (BE) and non-bioequivalent (NBE) formulations were donated by a Spanish pharmaceutical company. The excipients in the NBE formulations are croscarmellose sodium, magnesium stearate, microcrystalline cellulose, yellow iron oxide, colloidal silica, magnesium carbonate, and anhydrous lactose.

In the BE formulation, the excipients are magnesium stearate, microcrystalline cellulose, yellow iron oxide, povidone K30, calcium phosphate monobasic anhydrous, sodium starch glycolate, trometamol, and sodium phosphate dibasic dehydrate.

The company provided samples from the batches used in the BE study for tests and reference products.

The results obtained in their corresponding 2 × 2 crossover BE studies are reported in Table 1. The NBE formulation failed to show bioequivalence in C_max_.

### 2.3. Experimental Techniques

#### 2.3.1. Solubility Assays: Saturation Shake-Flask Method

To estimate pravastatin solubility, an excess of solid drug was added in buffer solutions pH 1.2, 4.5, and 6.8 at 37 °C. The solubility assays were performed according to the World Health Organization (WHO) guidelines protocols [10]. The equilibrium was reached in 8 h. A sample concentration was determined using high-performance liquid chromatography (HPLC) with ultraviolet (UV) detection.

#### 2.3.2. Lipophilicity Indexes: Partition Coefficients

Bulk phase partition coefficients (P) between *n*-octanol (analytical grade, Sigma-Aldrich, Barcelona, Spain) and phosphate buffer pH 6.80, 50 mM were determined for metoprolol and pravastatin.

The partition coefficient was calculated as the ratio between octanol concentration and the aqueous concentration. Three replicates were done to determine the average value.

Partition coefficients can be used as an index to provisionally classify compounds in terms of permeability [11]. Metoprolol was chosen as the reference compound for high permeability because its oral fraction absorbed is higher than 95%. Thus, drugs that exhibit partition coefficients and human intestinal permeability values lower than the value for metoprolol are considered low-permeability drugs.

#### 2.3.3. Permeability Assays: Cell Culture and Transport Studies

Caco-2 cells were grown in a polycarbonate membrane. To reach the confluence, 250,000 cells/cm^2^ were seeded in six Transwell plates and fasted for 19–22 days with Dubelcco’s Modified Eagle’s Media, with 1% l-glutamine, 10% fetal bovine serum, and 1% penicillin/streptomycin, at 37 °C temperature, 90% relative humidity, and 5% CO_2_.

The confluence of the cells was tested by measuring the transepithelial electrical resistance (TEER). Permeability studies were conducted with an orbital shaking (50 rpm) and with pH 7.0 in both chambers. Standard protocols were described and validated previously in our laboratory [12,13,14,15,16]. Four samples of 200 µL were taken and replaced with a fresh buffer from the receiver side at 15, 30, 45, and 90 min.

Pravastatin transport was studied in the solution at five concentrations (50, 100, 358 (highest single dose per tablet), 500, and 1000 μM). Permeability studies were performed in both directions: apical-to-basal (A-to-B) and basal-to-apical (B-to-A). The permeability value of pravastatin was compared with metoprolol, the high permeability reference compound.

The apparent permeability coefficient was calculated according the following equation:(1)$$C_{receiver,t}=\frac{Q_{total}}{V_{receiver}+V_{donor}}+\left(\left(C_{receiver,t-1}\cdot f\right)-\frac{Q_{total}}{V_{receiver}+V_{donor}}\right)\cdot e^{-P_{eff\;0,1}\cdot S\cdot \left(\frac{1}{V_{receiver}}+\frac{1}{V_{donor}}\right)\cdot \Delta t}$$
where *C_receiver,t_* is the drug concentration in the receiver chamber at time *t*, *Q_total_* is the total amount of drug in both chambers, *V_receiver_* and *V_donor_* are the volumes of each chamber, *C*_*receiver*,*t*−1_ is the drug concentration in the receiver chamber at the previous time, *f* is the sample replacement dilution factor, *S* is the surface area of the monolayer, Δ*t* is the time interval, and *P_eff_* is the permeability coefficient as was described by Mangas-Sanjuan et al. [14].

The permeability value of pravastatin 358 μM (highest single dose per tablet) was compared with the permeability value of the reference and test formulations at the same concentration of pravastatin. Experiments in the presence of the formulation excipients were done by dissolving a formulation tablet in 250 mL of buffer and filtrating the obtained dispersion to eliminate nonsoluble excipients.

#### 2.3.4. Disintegration

These assays were performed using a tablet disintegration tester (Hanson Research, Chatsworth, CA., USA) to measure the tablet disintegration time. According to the United States Pharmacopeia (USP) and European Pharmacopeia (Ph. Eur.) guidelines, the experiments were carried out in 800 mL of the media at 37 °C (*n* = 3).

The disintegration studies were performed in different media, simulating pH in the gastrointestinal human tract at pH 1.2, 4.5, and 6.8, and unbuffered water.

#### 2.3.5. Dissolution Assays

Drug release experiments were performed in 900 mL and 500 mL of pharmacopeia media (hydrochloric acid buffer/acetate buffer/phosphate buffer) at pH 1.2, 4.5, and 6.8, respectively, at 37 ± 0.5 °C and 50 rpm with USP 2 (Pharma-Test PT-DT70) [17]. Samples were taken at 5, 10, 15, 20, 30, 45, and 60 min, and the sample volume was replaced by a fresh preheated medium. Samples were immediately centrifuged (at 10,000 rpm for 10 min) and diluted (1:1) in methanol.

The dissolution profiles were compared by *f*_2_ (similarity factor) [18,19]. The *f*_2_ calculations were performed in Microsoft Excel (2016) [20].

### 2.4. Analysis of the Samples

The samples were analyzed by HPLC (Alliance-Waters 2695, Barcelona, Spain) using a Nova-Pak C18 column (4 μM, 3.9 × 150 mm) and UV detector (Waters 2487, Barcelona, Spain) at wavelengths of 238 nm. The flow-rate was 1.0 mL/min, and the mobile phase contained 50:40:10 methanol, water with 0.1% trifluoroacetic, and acetonitrile. The retention time of pravastatin was 3.2 min, and the limit of quantification was 0.02 µM. The analysis method fulfilled the linearity (*r* > 0.99), accuracy, and precision criteria (<5%).

The metoprolol HPLC method was published previously by our group [13,21].

### 2.5. Statistical Analysis

Results are shown as mean ± standard deviations. Statistical analysis of permeability values were two-tailed student t-tests, or analysis of variance (ANOVA) and Scheffé post hoc. The significance level was 0.05, and the software used was statistical package SPSS, V.20.00.

## 3. Results

The solubility pH profile for pravastatin at pH 1.2, 4.5, and 6.8 was 439.80 ± 17.42, 503.87 ± 24.20, and 479.58 ± 17.39 mg/mL (Figure 1). As expected considering its acidic nature, the lowest solubility was obtained at the lowest pH. The highest strength/dose of pravastatin (40 mg) would be soluble in 0.09 mL of water at pH 1.2. Dose number (Do) is 3.20 × 10^−4^. Therefore, pravastatin is a highly soluble drug. ANOVA test and Scheffé post hoc comparison detected differences in the solubility values at pH 1.2 versus the higher pHs, while no differences were detected between solubility values at pH 4.5 versus 6.8.

The *n*-Octanol partition coefficient was *P* = 0.70 ± 0.07 for pravastatin. This value was obtained with relation to a 65:35 *n*-octanol/phosphate buffer, with a pH 6.80, 50 mM. For metoprolol, the partition coefficient was *P* = 0.23 ± 0.05. According to these results, pravastatin could be provisionally classified as a high-permeability compound, but as Takagi, et al. [11] described previously, the existence of an active transport mechanism (absorptive or secretive) would bias the classification based purely on lipophilicity.

The permeability value of pravastatin was compared with a metoprolol (reference compound) value in order to classify pravastatin as a high- or low-permeability compound. Different concentrations of pravastatin were studied in order to characterize the transport mechanism of the drug across the intestinal membrane (Figure 2).

The analysis of variance (ANOVA) detected statistically significant differences among the permeability values obtained at different pravastatin concentrations (*p* = 0.0003).

Permeability values in the different formulations of pravastatin were also compared. The results are shown in Figure 3. The ANOVA and post hoc test showed statistically significant differences between the reference and NBE formulation (*p* < 0.05).

In Figure 4, disintegration times of pravastatin products are depicted with statistical differences between formulations. The disintegration endpoint of each sample is recorded as per the United States Pharmacopeia (USP) definition, in which no palpable form or outline of the sample is observed on the screen of the test apparatus or adhering to the lower surface of the disk.

Results of the dissolution tests in different buffers and volumes are summarized in Figure 5.

The reference and NBE products showed complete dissolution (>85%) in 15 min in the USP 2 at 50 rpm in 900 mL of buffered media at pH 1.2, 4.5, and 6.8. However, the BE product showed complete dissolution at 30 min. The same tests were conducted at 500 mL to verify whether a volume lower than 900 mL could detect differences in BCS class III formulations. However, the amount dissolved in these reduced volumes did not reach complete dissolution. Therefore, these dissolution tests are difficult to interpret.

## 4. Discussion

Based on the experimental results of solubility and permeability determinations, pravastatin is classified as a class III drug according with BCS, as its Dose number (Do) is less than 1 (≤3.20 × 10^−4^) at all the relevant pHs, and the permeability value of pravastatin was lower than that of metoprolol. Pravastatin is classified as a low-permeability drug, which is consistent with an oral bioavailability of 17% due to a low intestinal absorption (34% of dose administered) and a first-pass hepatic effect of 66% of the absorbed drug [22]. On the other hand, our experimental results are consistent with the existence of a secretion mechanism, as permeability increased at higher concentrations. The permeability value obtained in the presence of sodium azide that corresponded with the passive diffusion permeability confirms that not even at the highest pravastatin concentration tested was the efflux mechanism saturated. In addition, the permeability obtained in the presence of sodium azide, when any potential transporter contribution was nullified, was as high as metoprolol permeability, which is consistent with the experimentally estimated partition coefficient.

The observed changes in pravastatin permeability values in different formulations demonstrated that, in spite of the assumption of excipients being inert, actually some of them affect drug bioavailability, in particular for drugs with a carrier-mediated transport mechanism like pravastatin. Many excipients have shown its ability to inhibit the secretion activity of P-glycoprotein or MRP-2 transporters, increasing the permeability of the substrate drug [23,24,25,26]. Other authors have demonstrated that changes in paracellular route permeability are also affected by the presence of excipients [27,28,29]. The formulation with the largest permeability is the NBE formulation, which is consistent with the failed C_max_ due to supra-bioavailability.

Disintegration differences in pravastatin products were in line with the in vivo outcome for the NBE formulation, but there was no rank order correlation in the cases of the reference and the BE formulation, which showed marked differences in disintegration times that are not reflected in vivo. This might be attributed to the differences in permeability that are compensated for by the differences in disintegration/dissolution, since the BE formulation also exhibited higher permeability than the reference formulation, but this difference was compensated for by its slower disintegration/dissolution. In 500 mL of pH 6.8 buffer, the dissolution of the formulations showed the same trend as the observed in the in vivo results, and the *f*_2_ similarity factor indicated differences between the reference and NBE formulation. This may be because these dissolution conditions are predictive or simply a coincidence, since in these conditions complete dissolution was not achieved and the results are difficult to rely on.

Dissolution assays in the paddle apparatus at 50 rpm in 900 mL cannot detect differences in dissolution profiles of the reference and NBE formulations using classical buffers at pH 1.2, 4.5, and 6.8, because complete dissolution (>85%) was achieved in 15 min. However, in buffers of pH 1.2 and 6.8, it is evident that the dissolution was more rapid for the NBE formulation. This small difference, together with the large difference in permeability, seems to be the cause of the C_max_ failure, which was borderline, but the point estimate of the ratio test/reference was slightly >10% different, and the confidence interval did not include the 100% value. This difference in permeability highlights the relevance of requiring similar excipients for BCS class III drugs, as the high solubility could allow a similar in vitro dissolution, and even a similar in vivo dissolution, while still other factors may affect their oral fraction from being absorbed. For the BE product, the BCS-based biowaiver dissolution test in 900 mL would have led to a false negative result [3], i.e., differences observed in vitro while BE was observed in vivo. This possibility is not a problem from a regulatory point of view, since the companies always have the possibility to conduct an in vivo BE study, whereas the regulatory problem is to approve the NBE formulation based on in vitro dissolution profiles when the C_max_ is notably different and not able to show equivalence.

On the other hand, our dissolution results show that for these pravastatin products reduced volumes (500 mL) seem to be more discriminatory, i.e., both products are detected as nonsimilar at pH 1.2 and 4.5, and differences parallel or follow the same rank order correlation of the in vivo outcome at pH 6.8. However, these dissolution studies are difficult to interpret because complete dissolution is not reached and, therefore, the *f*_2_ similarity factor lacks any meaning. In fact, at pH 1.2 and 4.5, the amounts dissolved at 15 min, which is an approximation of the median gastric emptying time in the reference and the NBE formulations, can be considered similar. Therefore, with the simplistic assumption of gastric emptying after 15 min of residence in the stomach, the same amount would be released to the duodenum. The amounts dissolved at 15 min at pH 6.8 are different, but that pH is not the expected pH when the drug is emptied from the stomach. Therefore, if the C_max_ differences are due not only to the differences in permeability but also to differences in dissolution, it is evident that the usual volume of 900 mL is not adequate. Dissolution tests in 500 mL of buffer at 50 rpm were conducted to explore if volumes closer to the fluid volume in the gastrointestinal tract offer different results. However, the predictive power at pH 6.8 was not confirmed. For example, at pH 1.2 and 6.8, the *f*_2_ similarity factor was not able to conclude similarity because dissolution did not reach 100% at the end of the study but an asymptote. However, in relative terms, the NBE formulation and the reference exhibit similar amounts dissolved at 15 min (less than 10%). At pH 4.5, even the *f*_2_ similarity factors concludes with the NBE formulation and the reference.

## 5. Conclusions

The BCS-based biowaiver dissolution tests with the paddle apparatus at 50 rpm in pH 1.2, 4.5, and 6.8 in 900 mL media were not able to detect the in vivo C_max_ differences for pravastatin products. The different C_max_ seems to be the result of the combined effect of a higher permeability of the NBE formulation due to the excipients inhibition of the efflux transporters and a more rapid disintegration and dissolution. Those combined effects could not be detected with the current dissolution conditions in a volume of 900 mL and the criteria of similarity at 15 min, but the difference of the NBE formulation was observed at earlier sampling times (e.g., 5 min) and/or when the dissolution tests were conducted in 500 mL of dissolution media. Nevertheless, at 500 mL volume and pH 1.2 and 4.5, the BE formulation was also detected as nonsimilar.

## Figures and Tables

**Figure 1 pharmaceutics-11-00663-f001:**
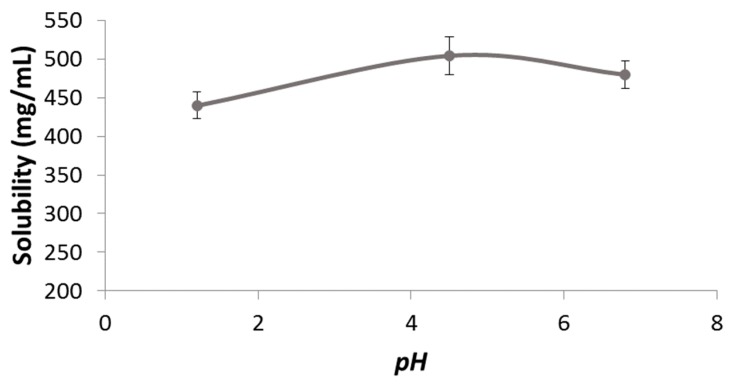
The pH solubility profile of pravastatin determined by the shake-flask method.

**Figure 2 pharmaceutics-11-00663-f002:**
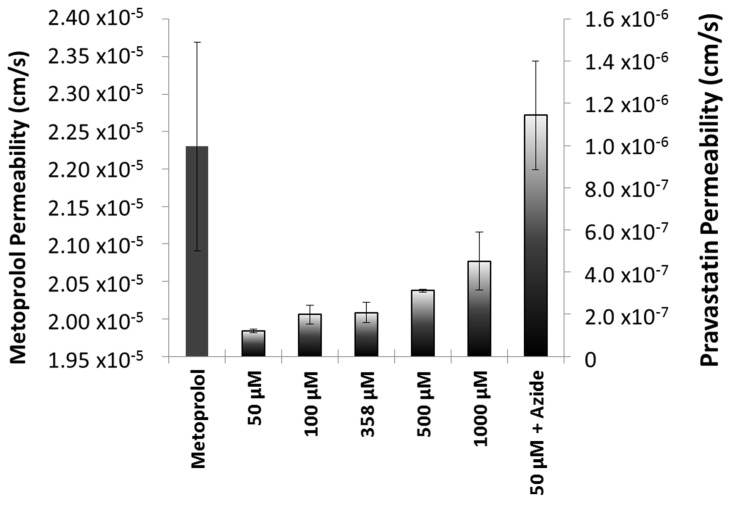
In vitro permeability values of pravastatin in Caco-2 cell monolayers at different drug concentrations. The highest clinical dose of pravastatin is equal to 358 µM (358 µM = 0.152 mg/mL, 100 µM = 0.04 mg/mL, 500 µM = 0.21 mg/mL, and 1000 µM = 0.42 mg/mL).

**Figure 3 pharmaceutics-11-00663-f003:**
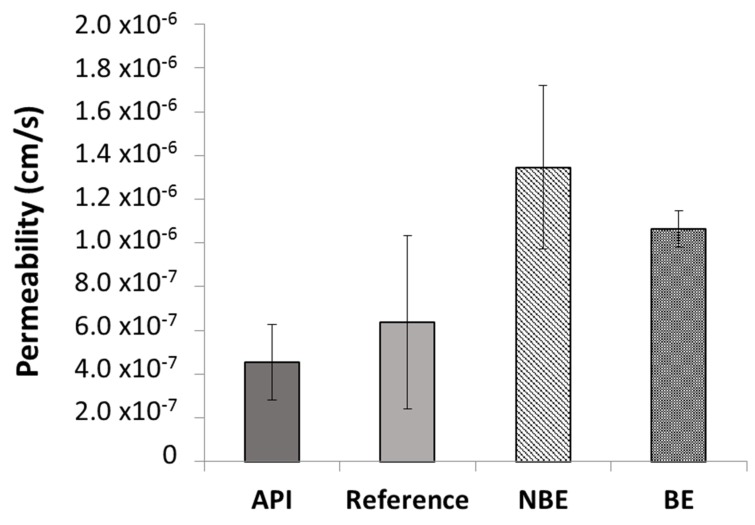
Pravastatin permeability values of the application programming interface (API) (358 µM), reference, non-bioequivalent (NBE), and bioequivalent (BE) formulations. Error bars represent the standard deviation.

**Figure 4 pharmaceutics-11-00663-f004:**
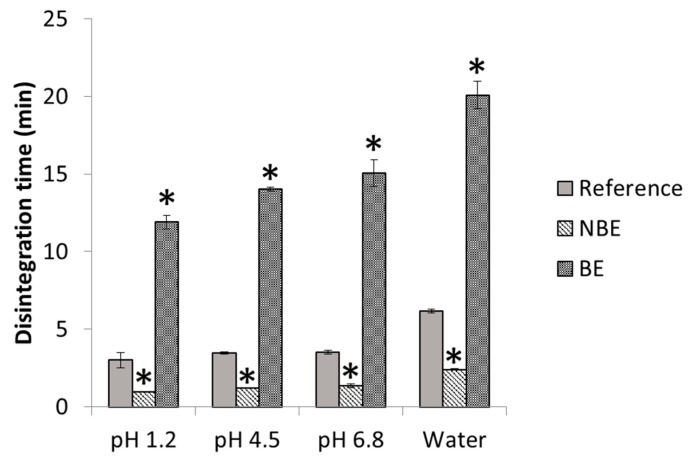
Disintegration times of various pravastatin products in different disintegration media. * Significant difference (*p* < 0.05).

**Figure 5 pharmaceutics-11-00663-f005:**
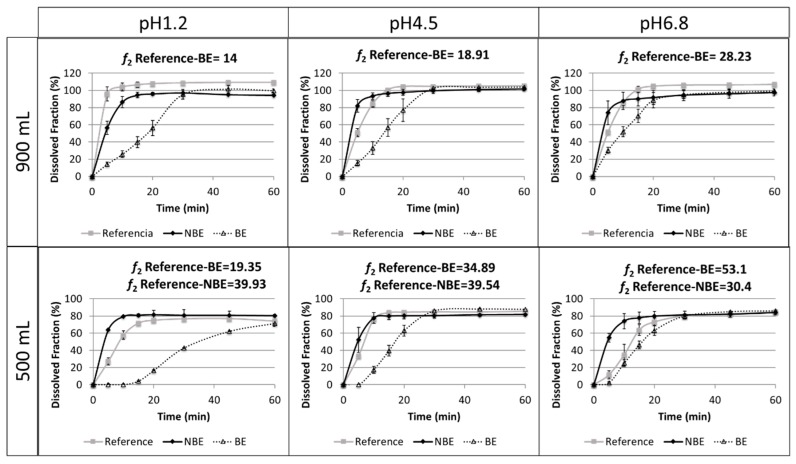
Dissolution profiles of three pravastatin formulations (reference product, non-bioequivalent formulation (NBE) and bioequivalent formulation (BE)) in the paddle apparatus at 50 rpm in buffered media at pH 1.2, 4.5, and 6.8 (*n* = 6).

**Table 1 pharmaceutics-11-00663-t001:** In vivo bioequivalence results of the test formulations.

Pharmacokinetic Parameter	Point Estimate and 90% Confidence Interval (%)	
	NBE Formulation	BE Formulation
C_max_	112.50 (100.20–126.30)	105.36 (95.66–116.04)
AUC_0-∞_	100.10 (92.50–107.50)	99.15 (92.90–105.82)

(*n* = 36 patients).

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
