# Peer review of "Investigation to Explain Bioequivalence Failure in Pravastatin Immediate-Release Products"

_pharmaceutics, 2019, doi:10.3390/pharmaceutics11120663_

Round 1

Reviewer 1 Report

Specific Comments:

 The authors wrote in the “1. Introduction” section:

57 outcome [5]. In the survey they found several drug products (including Pravastatin) that failed the

58 BE demonstration in spite of the adequate power of the BE study design (>80%) and the fact that the

59 drug products passed the QC dissolution test.

Comments:

The higher the study power the higher its ability to detect a difference if exists!.

The authors wrote in the “2.1. Compounds” section:

80 Pravastatin (MW=446.52 g/mol) was kindly provided by a pharmaceutical company.

Comments:

Pravastatin from which company (city and country)?

Under “2.3. Experimental Techniques” section:

The authors wrote:

118 Caco‐2 cells were grown in a policarbonate membrane.

Comments:

The spelling for “policarbonate” is not right.

For Disintegration and Dissolution Comments:

How many units were used in disintegration/dissolution experiments?

Under “3. Results” section:

164 The solubility‐pH profile for Pravastatin at pH 1.2, 4.5 and 6.8 was 439.80 ± 17.42, 503.87 ± 24.20

165 and 479.58 ± 17.39 mg/mL (Figure 1).

Comments:

The authors explained the solubility of the drug is lowest at acidic pH because of the acidic nature of the drug, but they did not explain the slight decrease in the solubility at the highest pH tested!.

Comments:

In figure 1, What do the error bars represent?

The authors wrote:

175 as high permeability compound but as Takagi et al [9]. described previously the existence of an active

Comments:

The authors need to remove the full stop after the reference [9].

Comments:

Error bars in figure 2 should be indicated as to what they represent.

Comments:

Error bars in figure 3 should be indicated as to what they represent.

The authors also need to indicate how they determined the endpoint of the disintegration.

Under “4. Discussion” section:

The authors wrote:

224 The observed changes in Pravastatin permeability values in different formulations

225 demonstrated that, in spite of the assumption of excipients being inert, actually some of them affect

226 drug bioavailability in particular for drugs with carrier mediated transport mechanism as

227 Pravastatin. Many excipients have shown its ability to inhibit secretion activity of P‐glycoprotein or

228 MRP‐2 transporter, increasing the permeability of the substrate drug. [20–23].

Comments:

The FDA indicates that “BCS class 3 drug products: Unlike for BCS class 1 products, for a biowaiver to be scientifically justified, BCS class 3 test drug product must contain the same excipients as the reference product.” Therefore the company producing the NBE product should have used the same excipients.

The authors wrote:

235 This might be due because the differences in permeability are compensated by the differences in

Comments:

The authors need to replace “due because” with “attributed to”.

The authors wrote:

238 In 500 ml of pH 6.8 buffer the dissolution of the formulations shown the same trend than the in vivo results.

Comments:

Replace “than the” with “to”.

Also, how can the authors explain that despite the drug being is highly soluble, the formulations failed to completely dissolve in 500 mL medium!.

The authors wrote:

239 results and the f2 similarity factor to detect differences between reference and NBE formulation. This

240 may be because these dissolution conditions are predictive or simply a coincidence, since in these

241 conditions complete dissolution was not achieved and the results are difficult to rely on.

Comments:

I could not understand. Please, rewrite this.

The authors wrote:

269 mL of buffer at 50 rpm were conducted to explore if volumes closer to the expected to be present in

270 the gastrointestinal tract offer different results.

Comments:

Replace “to the expected to be present” with “that”.

Comments:

Although the authors indicated (lines 257 to 275) that the dissolution in 500 mL is incomplete and difficult to interpret, the authors relied on the results to show correlation with the in vivo results!.

General Comments:

It is known that f2 from dissolution results is more discriminatory than in vivo bioequivalence study. How do you justify that the small differences in the dissolution had a significant effect on Cmax? Additionally, since more than 85% of the drug dissolves in less than 15 minutes (reference and NBE), this is a reason for biowaiver!.

How do you justify the significant difference between the in vitro dissolution of the reference and the BE formulation?

How clinically important is the difference in Cmax obtained for NBE formulation?

Did you test the potency of the products? You need to do that.

Do authors consider that it is possible in another in vivo bioequivalence study the NBE product would turn to be BE. According to statistics and especially with study power exceeding 80%, it is possible to find a difference between the reference and the NBE product because of chance (probability). The NBE product seems to be closer in dissolution to the reference than the BE product, the NBE product turns to be different from the reference.

The authors did not identify which excipient is causing the failed product (NBE).

The manuscript needs to be revised for spelling and grammar errors.

The authors did not indicate if they used software to calculate f2. They are encouraged to use the following reference:

Quality Attributes and In Vitro Bioequivalence of Different Brands of Amoxicillin Trihydrate Tablets. Pharmaceutics. 2017 May 20;9(2). pii: E18. doi: 10.3390/pharmaceutics9020018.

The following information from the FDA, could be useful to authors:  

For BCS class 3 drug products, the following should be demonstrated:

the drug substance is highly soluble • the drug product (test and reference) is very rapidly dissolving, and the test product formulation is qualitatively the same and quantitatively very similar (see section V.A.).

BCS class 3 drug products: Unlike for BCS class 1 products, for a biowaiver to be scientifically justified, BCS class 3 test drug product must contain the same excipients as the reference product. This is due to the concern that excipients can have a greater impact on absorption of low permeability drugs. The composition of the test product must be qualitatively the same (except for a different color, flavor, or preservative that could not affect the BA) and should be quantitatively very similar to the reference product. Quantitatively very similar includes the following allowable differences: • Changes in the technical grade of an excipient • Changes in excipients, expressed as percent (w/w) of the total formulation less than or equal to the following percent ranges: o Filler (± 10%) o Disintegrant, Starch (± 6%) o Disintegrant, Other (± 2%) o Binder (± 1%) o Lubricant, Calcium or Magnesium Stearate (± 0.5%) o Lubricant, Other (± 2%) o Glidant, Talc (± 2%) o Glidant, Other (± 0.2%) o Film Coat (± 2%) The total additive effect of all excipient changes should not be more than 10 percent.

Author Response

Comments:

The higher the study power the higher its ability to detect a difference if exists!.

Answer to comment

Thank you for this remark. We completely agree with the reviewer, the higher the power, the higher probability of detecting a difference in a standard statistical comparison in which the null hypothesis Ho is “difference not detected” and the alternative hypothesis, H1, that “a difference is detected”. Nevertheless in a BE study, a procedure called hypothesis inversion is used. In a Bioequivalence trial the null hypothesis, Ho, is the existence a predefined difference .i.e Bio-Inequivalence and the alternative hypothesis, H1 is “difference below the limit” i.e Bioequivalence. Thus in a BE test the power represents the probability of taking the right decision when accepting H1 that is Bioequivalence or “difference not detected”. Actually, increasing the power (by increasing sample size for instance), may help to conclude Bioequivalence in products with higher difference that may be declared Inequivalent with a lower sample size. (https://www.ncbi.nlm.nih.gov/pmc/articles/PMC4157693/)

To clarify that in the paper for the readers not familiar with the definition of hypothesis testing in BE trials the paragraph has been changed this way and two relevant references have been included.

In the survey they found several drug products (including Pravastatin) that failed the BE demonstration in spite of the adequate power of the BE study design (>80%) and the fact that the drug products passed the QC dissolution test. I.e even if the study had enough power to correctly conclude Bioequivalence (H1 alternative hypothesis), the products were found Inequivalent and null hypothesis (H0) of Inequivalence could not be rejected(PMID: 20857339, PMID: 25215170)

Reviewer comment

Pravastatin (MW=446.52 g/mol) was kindly provided by a pharmaceutical company. 

Comments:

Pravastatin from which company (city and country)?

DONE. Information add in the paper

Reviewer comment

The solubility‐pH profile for Pravastatin at pH 1.2, 4.5 and 6.8 was 439.80 ± 17.42, 503.87 ± 24.20, and 479.58 ± 17.39 mg/mL (Figure 1).

Comments:

The authors explained the solubility of the drug is lowest at acidic pH because of the acidic nature of the drug, but they did not explain the slight decrease in the solubility at the highest pH tested!.

Answer to comment

The values at pH 4.5 and 6.8 are not statistically different while solubility at pH 1.2 is different from those at higher pHs

To clarify this point the next sentence is added to the text.

ANOVA test and Scheffé post hoc comparison detected differences on the solubiliy values at pH 1.2 versus the higher pH’s while no differences were detected between solubility values at pH 4.5 versus 6.8.

Reviewer Comments:

Error bars in figure 3 should be indicated as to what they represent.

Done

Error bars represent standard deviation. The information has been included in the Figure legend

Reviewer Comments:

The authors also need to indicate how they determined the endpoint of the disintegration.

Done, clarify this information in the test

Endpoint was determined visually.

The disintegration endpoint of each sample was recorded as per the USP definition in which no palpable form or outline of the sample was observed on the screen of the test apparatus or adhering to the lower surface of the disk. 

Under “4. Discussion” section:

 Reviewer comment

The authors wrote:

224 The observed changes in Pravastatin permeability values in different formulations demonstrated that, in spite of the assumption of excipients being inert, actually some of them affect drug bioavailability in particular for drugs with carrier mediated transport mechanism as Pravastatin. Many excipients have shown its ability to inhibit secretion activity of P‐glycoprotein or MRP‐2 transporter, increasing the permeability of the substrate drug. [20–23].

 Comments:

The FDA indicates that “BCS class 3 drug products: Unlike for BCS class 1 products, for a biowaiver to be scientifically justified, BCS class 3 test drug product must contain the same excipients as the reference product.” Therefore the company producing the NBE product should have used the same excipients.

 Answer.

Thanks for pointing this out. That is correct. If a company wants to apply for a biowaiver of in vivo BE study and the API belongs to Class III then the same excipients as the reference product should be used. Nevertheless, in this case the company producing the NBE product never intended to request a biowaiver approach, thus they went through the in vivo BE demonstration and consequently they can use the excipients they deemed necessary and adequate.

Reviewer comment

The authors wrote:

235 This might be due because the differences in permeability are compensated by the differences in

Comments:

The authors need to replace “due because” with “attributed to”.

 Done

Change has been done accordingly in the text

Reviewer comment

The authors wrote:

In 500 ml of pH 6.8 buffer the dissolution of the formulations shown the same trend than the in vivo results.

Comments:

Replace “than the” with “to”.

Done

Grammar has been corrected. Thank for this correction.

Reviewer comment

Also, how can the authors explain that despite the drug being is highly soluble, the formulations failed to completely dissolve in 500 mL medium!.

 Answer

The formulation failed to dissolve completely in the timeframe of the dissolution study. Eventually, the product might dissolve completely. Actually, we agree with the reviewer that this is a surprising outcome not easy to explain but that clearly show the influence of some excipients on the API dissolution rate. Solubility at a given pH determines the dissolution rate gradient. On the other hand, other components of the dissolution rate equation might be highly influenced by excipients as the viscosity of the surrounding media. The formation of glassy and viscous aggregates may have interfered with the dissolution of pravastatin. An interesting aspect is that it happened for all the products not only the NBE one.

Reviewer comment

The authors wrote:

239 results and the f2 similarity factor to detect differences between reference and NBE formulation. This may be because these dissolution conditions are predictive or simply a coincidence, since in these conditions complete dissolution was not achieved and the results are difficult to rely on.

Comments:

I could not understand. Please, rewrite this.

Done

 The paragraph has been modified to

In 500 ml of pH 6.8 buffer the dissolution of the formulations shown the same trend as the observed in the in vivo results and the f2 similarity factor indicated differences between reference and NBE formulation. This may be because these dissolution conditions are predictive or simply a coincidence, since in these conditions complete dissolution was not achieved and the results, are difficult to rely on.

Reviewer comment

The authors wrote:

269 mL of buffer at 50 rpm were conducted to explore if volumes closer to the expected to be present in the gastrointestinal tract offer different results.

 Comments:

Replace “to the expected to be present” with “that”.

 Answer

Done

Paragraph has been changed

Reviewer comment

Although the authors indicated (lines 257 to 275) that the dissolution in 500 mL is incomplete and difficult to interpret, the authors relied on the results to show correlation with the in vivo results!.

Answer

Thanks for the remark. Clarify in the test that this conditions are not predictive

the difference of NBE formulation was observed at earlier sampling times (e.g. 5 minutes) and/or when the dissolution tests are conducted in 500 mL of dissolution media. Nevertheless, at 500 ml volume and pH 1.2 and 4.5 the BE formulation was also detected as non‐similar.

General Comments:

It is known that f2 from dissolution results is more discriminatory than in vivo bioequivalence study. How do you justify that the small differences in the dissolution had a significant effect on Cmax?

Answer

As the reviewer indicated for BCS class 1 and 3 drugs the in vitro “BCS dissolution conditions” (pH 1.2, 4.5 and 6.8 in 900 ml at 50 rpm) may be in some circumstances more discriminative then in vivo BE study. This is because for BCS class 1 an 3 there is not a “correlation” between in vitro dissolution rate and in vivo dissolution rate. The only assumption behind the BCS in vitro dissolution BE test is that if in vitro dissolution is fast and similar, nothing is opposed to assume in vivo dissolution will be also fast and similar and gastric emptying will be the limiting factor for absorption, for class 1 and membrane permeability for class 3. On the other hand, if the in vitro BCS dissolution test show differences it is not possible to predict the in vivo outcome as those differences may not be reflected in vivo.

The fact that small differences observed in vitro for a BCS class 3 drug are reflected in big differences in vivo in Cmax is the consequence of the potential effects of excipients on membrane permeability and transit times. For that reason the regulatory authorities request for BCS class 3 biowaivers, not only showing similarity on the in vitro tests but also containing similar excipients to avoid any risk of permeability or transit differences between products. This is one of the points in the present paper, showing that for a class 3 drug the excipient influence on permeability may cause absorption differences thus passing the in vitro test per se is a necessary but not sufficient condition to conclude BE.

Reviewer comment

Additionally, since more than 85% of the drug dissolves in less than 15 minutes (reference and NBE), this is a reason for biowaiver!.

Answer

A neccesary condition but not sufficient in class 3 drug products, as the excipients are not the same, thus being pravastatin a class 3, a biowaiver of the NBE product would have not been possible.

How do you justify the significant difference between the in vitro dissolution of the reference and the BE formulation?

Answer

By the fact that as mentioned by the reviewer for class 1 and 3 drugs the BCS dissolution conditions are many times overdiscriminative. Only similarity in combination with same excipients can be used to ensure similar in vivo outcome, but in vitro differences are meaningless for predicting in vivo outcome.

Reviewer comment

How clinically important is the difference in Cmax obtained for NBE formulation?

Answer

Based on a recent PK-PD meta-analysis of different statins the Cmax difference is expected to have a minor clinical relevance as in fact for pravastatin, there was no statistically significant difference in potency between b.i.d. and q.d. administration regimens (PMID: 24682029).

Reviewer comment

Did you test the potency of the products? You need to do that.

 Answer

Potency was checked and similar in both products which were within the accepted limits for the declared content.

Reviewer comment

Do authors consider that it is possible in another in vivo bioequivalence study the NBE product would turn to be BE. According to statistics and especially with study power exceeding 80%, it is possible to find a difference between the reference and the NBE product because of chance (probability). The NBE product seems to be closer in dissolution to the reference than the BE product, the NBE product turns to be different from the reference.

Answer

As we have concluded in the paper none of the tested the in vitro conditions are predictive of the in vivo outcome. On the other hand as we have explained previously the mentioned power 80% is the probability of correctly accepting H1 i.e. conclude BE. Of course the only risk which is fixed in the BE study is the alfa risk in this case meaning the risk of concluding BE when the products are truly BI and we can not rule out at 100% the possibility of the NoBE product passing the standard in another study. Nevertheless, the design and used sample size were the adequate for the purpose of protecting the consumer from BI products thus we are quite confident of the trial outcome.

Reviewer comment

The authors did not identify which excipient is causing the failed product (NBE).

 Answer

Thanks for the observation. It is correct, as we have tested the combined/overall influence of all the excipients in each formulation in pravastatin permeability. In consequence, the only conclusion that we can derive from the data is that the combination of excipients in NBe formulation affect permeability thus causing the BE failure. Identifying a particular excipient would have been speculative in the absence of experimental data

Reviewer comment

The manuscript needs to be revised for spelling and grammar errors.

 Answer

Thank you for the recommendation. We have revised accordingly the manuscript.

Reviewer comment

The authors did not indicate if they used software to calculate f2. They are encouraged to use the following reference:

 Quality Attributes and In Vitro Bioequivalence of Different Brands of Amoxicillin Trihydrate Tablets. Pharmaceutics. 2017 May 20;9(2). pii: E18. doi: 10.3390/pharmaceutics9020018.

Answer

F2 calculation can be easily implemented in an excel worksheet. We have used excel and indicated it in material and methods section

Add in the manuscript:

f2 calculations were performed in Microsoft excel (2016)  

Quality Attributes and In Vitro Bioequivalence of Different Brands of Amoxicillin Trihydrate Tablets. Pharmaceutics. 2017 May 20;9(2). pii: E18. doi: 10.3390/pharmaceutics9020018.

Reviewer comment 

The following information from the FDA, could be useful to authors:  

 For BCS class 3 drug products, the following should be demonstrated:

the drug substance is highly soluble • the drug product (test and reference) is very rapidly dissolving, and the test product formulation is qualitatively the same and quantitatively very similar (see section V.A.).

 BCS class 3 drug products: Unlike for BCS class 1 products, for a biowaiver to be scientifically justified, BCS class 3 test drug product must contain the same excipients as the reference product. This is due to the concern that excipients can have a greater impact on absorption of low permeability drugs. The composition of the test product must be qualitatively the same (except for a different color, flavor, or preservative that could not affect the BA) and should be quantitatively very similar to the reference product. Quantitatively very similar includes the following allowable differences: • Changes in the technical grade of an excipient • Changes in excipients, expressed as percent (w/w) of the total formulation less than or equal to the following percent ranges: o Filler (± 10%) o Disintegrant, Starch (± 6%) o Disintegrant, Other (± 2%) o Binder (± 1%) o Lubricant, Calcium or Magnesium Stearate (± 0.5%) o Lubricant, Other (± 2%) o Glidant, Talc (± 2%) o Glidant, Other (± 0.2%) o Film Coat (± 2%) The total additive effect of all excipient changes should not be more than 10 percent.

Answer

Thank you for the provided information that was already mentioned in the paper and the corresponding FDA and EMA guidance included as reference in line 251.

This difference in permeability highlights

249 the relevance of requiring similar excipients for BCS class III drugs, as the high solubility could allow

250 the similar in vitro dissolution and even a similar in vivo dissolution, while still other factors may

251 affect their oral fraction absorbed (references EMA guia de BE y guia FDA de biowaivers)

Reviewer 2 Report

The manuscript reports on the important aspect of bioequivalence evaluation related to investigation of bioequivalence failure for immediate release products. Pravastatine reference products and two generic products – one with in vivo proven bioequivalence and one which exhibited “suprabioavailability” in vivo were tested with respect to tablet disintegration and drug dissolution in media with different pH values. The results obtained indicate that in vitro dissolution was not predictive on the in vivo behaviour of the investigated formulations.

The manuscript is generally well written, however there are some issues which should be further clarified.

Investigated products

It is not clear if comparative in vitro study has been performed with the same product batches which were tested in vivo, as it is of outmost importance to compare the in vitro and in vivo data for the same batch. If this is not the case, comparative in vitro data should be presented for at least three different batches of each product in order to document uniformity and reproducibility of data.

Permeability study

It has been stated that “pravastatin transport was studied in solution at five concentrations in the presence of each excipient of the formulation”. More detailed information on the amount of excipients used and how they relate to formulation composition and drug:excipient ratio should be provided. Addition of sodium azide was not mentioned in the Experimental techniques section.

It appears on Figure 3 that API reference permeability refers to 1000 mM sample. It is previously stated that highest clinical dose of pravastatine equals to 358 mM. Please describe how permeability of the ‘reference’, NBE and BE samples have been determined.

Dissolution study

The number of tablets tested should be provided.

The reason for high SD values observed for the reference and NBE product in 500 ml media pH 1.2 should be discussed (samples at the plateau phase, at 20, 30 and 45 min). Also, the reason for incomplete drug dissolution in 500 ml media, although the plateau is reached in majority of cases at the same time point as in 900 ml media.

Author Response

Reviewer 2

Investigated products

It is not clear if comparative in vitro study has been performed with the same product batches which were tested in vivo, as it is of outmost importance to compare the in vitro and in vivo data for the same batch. If this is not the case, comparative in vitro data should be presented for at least three different batches of each product in order to document uniformity and reproducibility of data.

Answer

The next sentence has been included in material ad methods

The company provided samples form the batches used in the BE study for test and reference products.  

Permeability study

It has been stated that “pravastatin transport was studied in solution at five concentrations in the presence of each excipient of the formulation”. More detailed information on the amount of excipients used and how they relate to formulation composition and drug:excipient ratio should be provided. Addition of sodium azide was not mentioned in the Experimental techniques section.

Answer:

This statement is in lines 127 and 128 and it has been corrected

Pravastatin transport was studied in solution at five concentrations (50, 100, 358 (highest single

128 dose per tablet), 500 and 1000 µM) and in the presence of the excipients of each of the formulations

This was a mistake. In the final version we stated in lines 178-181:

The permeability value of Pravastatin was compared with Metoprolol (reference

compound) value in order to classify Pravastatin as high or low permeability compound. Different

concentrations of Pravastatin were studied in order to characterize the transport mechanism of the

drug across the intestinal membrane. (Figure 2)

It appears on Figure 3 that API reference permeability refers to 1000 mM sample. It is previously stated that highest clinical dose of pravastatine equals to 358 mM. Please describe how permeability of the ‘reference’, NBE and BE samples have been determined.

Answer

Figure 3 data was obtained at 358µM.

It has been indicated in the text by including this paragraph in M&M section

Permeability value of Pravastitin 358 mM (highest single dose per tablet) was compared with permeability value of reference and test formulations at same concentration of Pravastatin. Experiments in the presence of the formulations excipients were done by dissolving a formulation tablet in 250 mL of buffer and filtrating the obtained dispersion to eliminated non-soluble excipients

Dissolution study

The number of tablets tested should be provided.

The reason for high SD values observed for the reference and NBE product in 500 ml media pH 1.2 should be discussed (samples at the plateau phase, at 20, 30 and 45 min). Also, the reason for incomplete drug dissolution in 500 ml media, although the plateau is reached in majority of cases at the same time point as in 900 ml media.

Answer

number of tablets tested has beed added in the manuscript (n=6)

We do not have a clear explanation for the fact of the incomplete dissolution in 500 mL. We have already mentioned this fact as a limitation to the interpretation of 500 mL data.

It may be the still slower disintegration in these conditions and the different hydrodynamic conditions with a more pronounced coning effect with the lower volume that slow down the dissolution rate which could eventually be complete if the test would have lasted more time. We can not rule out either some influence of the excipients in the api solubility which we have observed for other high solubility drugs.

The reason for high SD values observed for the reference and NBE product in 500 ml media pH 1.2 should be discussed (samples at the plateau phase, at 20, 30 and 45 min)

Thanks for this comment. It was a mistake. Graphs has been changed.

Round 2

Reviewer 1 Report

No comments.

Reviewer 2 Report

The authors addressed all comments sufficiently. Careful text editing of the final revision should be performed.